# IL-26 in the Lung and Its Role in COPD Inflammation

**DOI:** 10.3390/jpm12101685

**Published:** 2022-10-09

**Authors:** Konstantinos Bartziokas, Evangelia Fouka, Stelios Loukides, Paschalis Steiropoulos, Petros Bakakos, Andriana I. Papaioannou

**Affiliations:** 1Pulmonologist, 42100 Trikala, Greece; 2Respiratory Medicine Department, George Papanikolaou Hospital, University of Thessaloniki, 57010 Thessaloniki, Greece; 32nd Respiratory Medicine Department, “Attikon” University Hospital, National and Kapodistrian University of Athens, 12462 Athens, Greece; 4Department of Respiratory Medicine, Medical School, University General Hospital Dragana, Democritus University of Thrace, 68100 Alexandroupolis, Greece; 51st University Department of Respiratory Medicine, “Sotiria” Hospital, National and Kapodistrian University of Athens, 15772 Athens, Greece

**Keywords:** IL-26, COPD, lung inflammation

## Abstract

IL-26 is a cytokine expressed by infiltrating pro-inflammatory IL-17-producing T cells in the tissues of patients with chronic lung inflammation. IL-26 induces the chemotactic response of human neutrophils to bacteria and other inflammatory stimuli. In recent years, the innovative properties of IL-26 have been described. Studies have shown that, as DNA is released from damaged cells, it binds to IL-26, which plays the role of a carrier molecule for extracellular DNA, further contributing to its binding to the site of inflammation. This mechanism of action indicates that IL-26 may serve both as a driver as well as a stimulus of the inflammatory process, leading to the installation of a noxious amplification loop and, eventually, persistent inflammation. IL-26 also demonstrates direct antimicrobial effects derived from its capability to create pores and disrupt bacterial membranes, as indicated by the presence of membrane blebs on the surface of the bacteria and cytosolic leakage pores in bacterial walls, produced in response to microbial stimuli in human airways by several different immune and structural cells. Surprisingly, while this particular cytokine induces the gathering of neutrophils in areas of infection, it also exhibits inhibitory and pro-inflammatory effects on airway epithelial and immune cells. These remarkable effects underline the necessity of a better understating of its biological behavior and its role in the pathophysiology and disease burden in several smoking-related airway inflammatory disorders, such as Chronic Obstructive Pulmonary Disease (COPD) and chronic bronchitis. In this review, we aim to discuss the current role of IL-26 in the lung, with an emphasis on systemic inflammation in patients suffering from COPD and chronic bronchitis.

## 1. Introduction

The term “cytokine” is a compound that derives from the combination of two Greek words—“cyto”, which means cell, and “kinos”, meaning activity. Cytokines represent a large family of signaling molecules that mediate cell-to-cell communication in immune responses and provoke the migration of cells to the sites of infection, injury, and inflammation [1]. As cytokines also regulate both innate and adaptive immune responses [2], the expression “immunomodulating agents” is often used against the term cytokines.

Currently, no unified classification system for cytokines exists, and their categorization is based on either the cell types from which they are produced or their functional properties. Consistent with a classification based on the similarity of receptor compartments and structures, cytokines are divided into four major groups: (i) type I cytokines (comprised of interleukins (IL)-2, IL-3, IL-4, IL-5, IL-6, IL-7, IL-8, IL-9, IL-11, IL-12, IL-13, and IL-15), (ii) type II cytokines (including interferons (IFNs) and the IL-10 superfamily), (iii) tumor necrosis factor (TNF) superfamily (TNFs, Fas ligands, etc.), and (iv) immunoglobulin (Ig) superfamily (IL-1/IL-18) [3,4].

The IL-10 superfamily, which belongs to the type II group of cytokines, has nine members, namely, IL-10, IL-19, IL-20, IL-22, IL-24, IL-26, IL-28A, IL-28B, and IL-29, and, except for IL-10, which applies to immunomodulatory functions, all other members are pro-inflammatory cytokines that interfere with inflammation [5]. Intriguingly, the IL-10-related cytokine IL-26 is largely expressed in human airways [6], and alterations in its expression are associated with decreased lung function and reductions in markers of neutrophilic inflammation in Chronic Obstructive Pulmonary Disease (COPD) [7]. In this review, we aim to discuss the current role of IL-26 in the lung, with an emphasis on systemic inflammation in patients suffering from COPD and chronic bronchitis.

## 2. Expression of IL-26 in the Lung

IL-26 is expressed in both immune and non-immune cells. Activated CD4+ T helper (Th)1 and Th17 cells are the main secretors of IL-26, whereas CD4+ regulatory T cells (Tregs) and Th2 cells have almost no expression of this cytokine [8]. Moreover, naive CD4+ T cells also express IL-26, although at lower levels compared to memory CD4+ T cells [8]. Specifically, IL-26 is expressed by infiltrating pro-inflammatory IL-17-producing T cells in the tissues of patients with chronic lung inflammation [6,9]. Therefore, IL-26 can be considered as a marker of high Th17 cell differentiation [10]. Furthermore, Ohnuma and colleagues outlined that CD26+ CD4+ T cells constitute a significant source of IL-26 in a mouse model of graft-versus-host disease (GVHD) [11]. Eventually, cytotoxic CD8+ T cells and IL-22-expressing natural killer (NK) cells present in mucosa-associated lymphoid tissues, also called NK22 or innate lymphoid cell 3 (ILC3), have also been reported to produce IL-26 [12].

However, the exact nature of the signals required for the expression of IL-26 by human T cells has not yet been clarified. IL-1β and IL-23, two essential cytokines for the production of human Th17 cells, actuate IL-26 production by CD4+ T lymphocytes [13,14], and, conversely, IL-26-stimulated CD4+ T cells secrete IL-17 and IL-23 [15]. This loop of cytokine induction may participate in the maintenance of the pro-inflammatory phenotype of infiltrating memory T cells within IL-26-containing inflamed tissues [16].

The expression of IL-26 in other immune cells, such as monocytes and macrophages, has also been observed, but at lower levels compared with T cells, while its expression is upregulated upon stimulation by lipopolysaccharides (LPS) and IFN-γ [12]. In addition to alveolar macrophages, it has been confirmed that primary human lung fibroblasts (HLFs) produce IL-26 in response to endotoxin, contributing to the release of the neutrophil-mobilization cytokines IL-6 and IL-8, both constitutively and after exposure [17]. In this study, the investigators discovered that the stimulation of HLF via endotoxin generates an increased and sustained phosphorylation of NF-κB, MAP kinases p38, JNK1-3, and extracellular signal-regulated kinase (ERK)1/2. In contrast, the glucocorticoid hydrocortisone significantly inhibited the endotoxin-induced release of IL-26, IL-6, and IL-8, an effect paralleled by the decreased phosphorylation of ERK1/2, p38, and NF-κB. Likewise, tiotropium, but not aclidinium, both well-known muscarinic receptor antagonists, resulted in a minor inhibition of the endotoxin-induced release of interleukins 26 and 8, paralleled by a decreased phosphorylation of NF-κB. Finally, the short-acting, selective beta2-adrenergic receptor agonist salbutamol contributed to a modest inhibition of the endotoxin-induced release of IL-26 and IL-8, paralleled by a decreased phosphorylation of JNK1-3, NF-κB, and p38. Almost identical pharmacological effects were noticed for the constitutive release of IL-26. Consequently, since the endotoxin-induced phosphorylation of NF-κB and p38, c-Jun N-terminal kinase (JNK)1-3, and ERK1/2 in HLF are matched by an increased release of IL-26, these intracellular signaling pathways are implicated in the endotoxin-induced release of IL-26 in HLF. Therefore, given their abundance in the lungs, HLFs may represent an essential source of IL-26 in human airways.

Among non-immune cells, IL-26 expression was also established in human primary bronchial epithelial cells in response to in vitro viral stimulation [18]. In this study, stimulation with the viral-related TLR3 agonist poly-IC increased the phosphorylation of the identical intracellular signaling molecules, matched by an increased release of IL-26 [18]. In contrast, in the same study, the selective inhibition of these intracellular signaling compounds resulted in the reduced release of IL-26 protein [18]. Fibroblast-like synoviocytes, macrophage-like synoviocytes, and primary smooth muscle cells can also produce IL-26 after exposure to inflammatory cytokines, including TNF-α, IL- 17A, and IL-1β [16,19].

## 3. Mechanism of Action

IL-26 induces the chemotactic response of human neutrophils to bacteria and other inflammatory stimuli [6,20], thus implying a crucial function exerted in a critical immune barrier. IL-26 exerts these properties after binding to a heterodimeric receptor composed of IL-10R2 and IL-20R1; the defined binding site of IL-26 to this receptor complex is located only on IL-20R1, but the whole complex is required for signaling [21]. Accordingly, the anti-IL10R2 antibody blocks IL-26-induced signal transduction, indicating direct evidence that IL-10R2 is essential to assembling a functional IL-26 receptor complex [22,23]. Although IL-20R1 is restricted to epithelial and some myeloid cells, IL-10R2 is ubiquitously expressed [6,24]. Bao and colleagues have also shown that IL-26 induces the endotoxin-induced mobilization of neutrophils toward the bronchoalveolar space in a mouse model in vivo, which expresses the IL-26 receptor complex (IL-10R2 and IL20R1) [25].

Cytokines are considered to play a critical orchestrating role in the chronic inflammation observed in various inflammatory disorders, such as COPD, and many of them signal through the Janus kinase (JAK) signal transducer and the activator of signal transducer and activator of transcription (STAT) pathways and are produced as a consequence of JAK-STAT pathway signaling [26]. IL-26 has been found to induce JAK1 and tyrosine kinase (TyK)2 signaling, leading to STAT1 and STAT3 phosphorylation and activation [21,23]. Strong evidence suggests a leading position for the JAK-STAT pathway in the more severe stages of COPD, specifically STAT1 and STAT3, which are targets for the novel COPD therapeutics [27].

In recent years, increasingly innovative properties of IL-26 have come to the fore, which are attributed to its non-conventional cationic and amphipathic features. As DNA is released from damaged cells, it binds to IL-26, which plays the role of a carrier molecule for extracellular DNA, further contributing to its binding to the site of inflammation [28]. This mechanism of action indicates that IL-26 may serve both as a driver as well as a stimulus of the inflammatory process, leading to the installation of a noxious amplification loop and, eventually, persistent inflammation, acting as a crucial moderator of local immunity and inflammation [12].

IL-26 also demonstrates direct antimicrobial effects. The antimicrobial properties of IL-26 are derived from its capability to create pores and disrupt bacterial membranes, as indicated by the presence of membrane blebs on the surface of the bacteria and cytosolic leakage pores in bacterial walls [29]. IL-26 eliminates or inhibits the proliferation of numerous Gram-positive (*Staphylococcus aureus*) and Gram-negative bacteria (*Klebsiella pneumoniae, Escherichia coli, Pseudomonas aeruginosa*) at concentrations such as the antimicrobial molecule LL-37. Recently, Woetmann et al. verified this particular effect of IL-26, demonstrating that this specific interleukin activates *Staphylococcus aureus* death and inhibits biofilm formation [30]. IL-26 also exerts indirect antimicrobial activity by inducing the secretion of type I and type II INFs from NK cells, monocytes, and plasmacytoid dendritic cells (pDC) [28,31]. Notably, for pDC activation, the presence of IL-26 is required, since the killing of *Pseudomonas aeruginosa* alone cannot induce the production of IFNα [29]. All of these biological functions of IL-26 classify it as a membrane-active antimicrobial peptide [32].

Interestingly, in an animal model of transgenic mice with transplant-related obliterative bronchiolitis, an increase in IL-26 expression and collagen deposition, inhibited by neutralizing anti-IL 26 antibodies, was demonstrated 4 weeks post-transplantation [11]. These findings suggest a possible association of long-term IL-26 upregulation with airway remodeling and an additional potent benefit of the therapeutic targeting of IL-26.

## 4. The Role of IL-26 in COPD and Chronic Bronchitis

COPD is characterized by chronic inflammation that causes structural changes and narrowing of the small airways, leading to airflow obstruction in the lungs [33]. Chronic inflammation in the lower parts of the respiratory tract and the lungs of patients with stable COPD has been associated with the increased expression of several cytokines, such as TNF-α, INF-γ, IL-1β, IL-6, IL-17, IL-18, IL-32, thymic stromal lymphopoietin (TSLP), and growth factors, such as transforming growth factor-beta (TGF-β) [34]. Given the fact that Th17 cells exert an antibacterial effect in pulmonary tissue, participating in the accumulation of macrophages and neutrophils during the inflammatory process [35], and as IL-26 involvement has been associated with antibacterial responses in the airways, the evaluation of the role of IL-26 in the evolution and persistence of inflammatory activity in various disorders such as COPD makes it of considerable interest to researchers [36].

In a cohort of eighty-three patients with stable COPD, a significant increase in IL-26 in induced sputum was observed compared to healthy subjects [37]. In this study, IL-26 levels in sputum were shown to correlate positively with Body Mass Index (BMI), C-Reactive Protein (CRP), and leptin. At the same time, negative correlations with Forced expiratory volume in 1 s (FEV_1_) and FEV_1_/FVC (forced vital capacity) were also reported, suggesting that this specific cytokine can be considered a prospective marker for detecting the degree of inflammation in the lung tissue of COPD patients.

Cigarette smoking remains the leading cause of COPD in the industrialized world and a principal causative factor for a common condition called chronic bronchitis [38]. Long-term tobacco smokers with COPD or chronic bronchitis exhibit an extravagant accumulation of neutrophils in the airways, a type of inflammation that exhibits poor responses to traditional therapy [39]. Che et al. detected increased levels of extracellular IL-26 in the airways by analyzing bronchial wash (BW), bronchoalveolar lavage (BAL), and induced sputum samples from long-term smokers with and without COPD and chronic bronchitis [40]. In human alveolar macrophages in vitro, the exposure to water-soluble tobacco smoke components enhanced the IL-26 gene and protein and increased the gene expression of the IL-26 receptor complex (IL10R2 and IL20R1) and nuclear factor κ-B (NF-κB), an established regulator of the production of IL-26 [40]. Similarly, in the same study, extracellular IL-26 in BAL samples was found to correlate with lung function parameters (FEV_1_% predicted and FEV_1_/FVC%), tobacco load (pack-years), and markers of neutrophil aggregation. Likewise, extracellular IL-26 levels in long-term smokers were further increased in the induced sputum in patients with exacerbations of COPD compared to steady-state patients, in BAL samples from patients with chronic bronchitis, and in the induced sputum and BW samples in subjects with chronic colonization by pathogenic bacteria [40]. Interestingly, a gradual and statistically significant increase in IL-26 levels in induced sputum was detected in the period prior to an exacerbation, occurring 17 (14–23) days before the clinical event, indicating a potential role of IL-26 as a biomarker of upcoming exacerbations [7,40].

COPD and obesity are two reciprocally aggravating disorders sharing a common pathogenetic mechanism: chronic systemic inflammation [41,42,43]. Inflammation in COPD is located in lung tissue and, more specifically, in bronchial tubes [44], whereas, in obesity, it is found in adipose tissue [45]. Leptin is a hormone predominantly made by adipocytes, revealing pleiotropic effects in both hereditary and acquired immunity [46]. Studies of the last decade indicate that, except for adipocytes, leptin is also secreted by numerous epithelial cells (intestinal, gastric, mammary, bronchial epithelial cells, and type II pneumocytes) [47]. More specifically, in ex-smokers with or without severe COPD, leptin was detected in induced sputum, BAL, and proximal and peripheral lung tissue biopsies, while its expression has been reported to be increased in alveolar macrophages and bronchial epithelial cells [48]. In addition, leptin levels were found to be positively correlated with BMI and CRP and negatively correlated with lung function parameters (FEV_1_ and FEV_1_/FVC) [37]. In a trial aiming to evaluate lifestyle interventions in obese patients with COPD, the implementation of individually developed therapeutic strategies incorporating nutritional improvement and regular physical activity led to a reduction in IL-26 and CRP levels compared to standard inhaled therapy, a finding indicating the importance of IL-26 in obese COPD patients [49].

## 5. Conclusions

IL-26, a cytokine belonging to the IL-10 superfamily, is produced in response to microbial stimuli in human airways by several different immune and structural cells. It modulates the host defense mechanisms of human airways and exerts direct and indirect antimicrobial properties. Surprisingly, while this particular cytokine induces the gathering of neutrophils in areas of infection, it also exhibits inhibitory and pro-inflammatory effects on airway epithelial and immune cells. Moreover, IL-26 upregulation has been found to be associated with reduced lung function, enhanced systemic inflammation, and upcoming exacerbations, suggesting a potent role in disease monitoring while serving as a biomarker of disease progression and exacerbation risk in patients with COPD.

These remarkable effects underline the necessity of a better understating of the biological behavior of IL-26 and its role in the pathophysiology and disease burden in several smoking-related airway inflammatory disorders, such as COPD and chronic bronchitis. However, we still have a long way to go before we fully understand whether IL-26 possesses any clinical utility as a potent biomarker for diagnosis and monitoring and, more importantly, as a therapeutic target of one or more specific endotypes of COPD—particularly, those related with Th17-related neutrophilic airway inflammation.

## Data Availability

Not applicable.

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
