# Peer review of "IL-26 in the Lung and Its Role in COPD Inflammation"

_jpm, 2022, doi:10.3390/jpm12101685_

Round 1

Reviewer 1 Report

This is a well-written paper with a good structure and valuable information on the role of IL-26. However, I ask authors to revise some areas that require further detailed description for the role of IL-26 in COPD. Thank you for giving me the opportunity to review this manuscript.  

Major

1. COPD during stable clinical conditions is different than acute exacerbation of COPD. I recommend the authors should describe the role IL-26 in acute exacerbation of COPD as in the papers by Che KF et al and Cardenas EI et al.(Che KF, Tufvesson E, Tengvall S, et al. The neutrophil-mobilizing cytokine interleukin-26 in the airways of long-term tobacco smokers. Clin Sci. 2018;132(9):959–983, Eduardo I. Cardenas, Karlhans Fru Che, Jon R. Konradsen, Aihua Bao & Anders Lindén Expert Rev Respir Med. 2022 Mar;16(3):293-301.)

2. I recommend authors should describe the role of IL-26 regarding the possibility of therapeutic target in COPD or whether it can work as a biomarker for disease progression of COPD.

Minor

1. The sentence in 180 -183 ( Likewise, extracellular IL-26 levels in long-term smokers were further increased in induced sputum in patients with exacerbations of COPD, in BAL samples from patients with chronic bronchitis, and in induced sputum and BW samples in subjects with chronic colonization by pathogenic bacteria) requires references.

2. The sentence in 99-101 ( In this study, stimulation with the viral-related TLR3 agonists poly-IC, increased phosphorylation of the identical intracellular signaling molecules, matched by an increased release of IL-26.) needs references.

Author Response

Major

  1. COPD during stable clinical conditions is different than acute exacerbation of COPD. I recommend the authors should describe the role IL-26 in acute exacerbation of COPD as in the papers by Che KF et al and Cardenas EI et al. (Che KF, Tufvesson E, Tengvall S, et al. The neutrophil-mobilizing cytokine interleukin-26 in the airways of long-term tobacco smokers. Clin Sci. 2018;132(9):959–983, Eduardo I. Cardenas, Karlhans Fru Che, Jon R. Konradsen, Aihua Bao & Anders Lindén Expert Rev Respir Med. 2022 Mar;16(3):293-301.)

We thank the reviewer for his/her comment. A paragraph has been added in the manuscript as follows (page 10) as follows:

“Similarly, in the same study, extracellular IL-26 in BAL samples was found to correlate with lung function parameters (FEV1% predicted and FEV1/FVC%), tobacco load (pack-years), and markers of neutrophil aggregation. Likewise, extracellular IL-26 levels in long-term smokers were further increased in induced sputum in patients with exacerbations of COPD compared to steady-state, in BAL samples from patients with chronic bronchitis, and in induced sputum and BW samples in subjects with chronic colonization by pathogenic bacteria. Interestingly, a gradual and statistically significant increase in IL-26 levels in induced sputum was detected in the period prior to an exacerbation, occurring 17 (14-23) days preceding the clinical event, indicating a potential role of IL-26 as a biomarker of upcoming exacerbations [7, 40].

  1. I recommend authors should describe the role of IL-26 regarding the possibility of therapeutic target in COPD or whether it can work as a biomarker for disease progression of COPD.

We thank the reviewer for his/her comment. A paragraph has been added in the manuscript (page 8) as follows:

’Interestingly, in an animal model of transgenic mice with transplant-related obliterative bronchiolitis, an increase in IL-26 expression and collagen deposition, inhibited by neutralizing anti-IL 26 antibodies, was demonstrated 4 weeks post-transplantation [11]. These findings suggest a possible association of long-term IL-26 upregulation with airway remodeling and an additional potent benefit of therapeutic targeting of IL-26.”

and in the conclusion section as follows:

“Moreover, IL-26 upregulation has been found to be associated with reduced lung function, enhanced systemic inflammation and upcoming exacerbations, suggesting a potent role in disease monitoring, while serving as a biomarker of disease progression and exacerbation risk in patients with COPD.”

Minor

  1. The sentence in 180 -183 (Likewise, extracellular IL-26 levels in long-term smokers were further increased in induced sputum in patients with exacerbations of COPD, in BAL samples from patients with chronic bronchitis, and in induced sputum and BW samples in subjects with chronic colonization by pathogenic bacteria) requires references.

We thank the reviewer for his/her comment. A reference has been added as requested and now appears as reference 40

  1. The sentence in 99-101 (In this study, stimulation with the viral-related TLR3 agonists poly-IC, increased phosphorylation of the identical intracellular signaling molecules, matched by an increased release of IL-26.) needs references.

We thank the reviewer for his/her comment. A reference has been added as requested and now appears as reference 18

Reviewer 2 Report

Inflammatory pathways in COPD are addressed by the research.

The topic is relevant and interesting. I didn’t find  similar review about IL-26.

Good integrative approach if we look it as a review article!

The paper is well written. The text is clear and easy to read.

Conclusions consistent with the evidence and arguments.

Comprehensive review article.

Author Response

Inflammatory pathways in COPD are addressed by the research.

The topic is relevant and interesting. I didn’t find similar review about IL-26.

Good integrative approach if we look it as a review article!

We thank the reviewer for his commen

Round 2

Reviewer 1 Report

This paper is well revised, so I have no further comment.